# Using Large Ensembles of Control Variates for Variational Inference

**Tomas  Geffner**
College of Information and Computer Science
University of Massachusetts
Amherst, MA 01003
tgeffner@cs.umass.edu

**Justin  Domke**
College of Information and Computer Science
University of Massachusetts
Amherst, MA 01003
domke@cs.umass.edu

## Abstract

Variational inference is increasingly being addressed with stochastic optimization. In this setting, the gradient's variance plays a crucial role in the optimization procedure, since high variance gradients lead to poor convergence. A popular approach used to reduce gradient's variance involves the use of control variates. Despite the good results obtained, control variates developed for variational inference are typically looked at in isolation. In this paper we clarify the large number of control variates that are available by giving a systematic view of how they are derived. We also present a Bayesian risk minimization framework in which the quality of a procedure for combining control variates is quantified by its effect on optimization convergence rates, which leads to a very simple combination rule. Results show that combining a large number of control variates this way significantly improves the convergence of inference over using the typical gradient estimators or a reduced number of control variates.

## 1   Introduction

Variational Inference (VI) [29, 2, 11] is a framework for approximate probabilistic inference. It has been successfully applied in several areas including topic modeling [3, 21], generative models [13, 5, 22], reinforcement learning [6], and parsing [15], among others. Recently, VI has been able to address a wider range of problems by adopting a "black box" [25] view based on only evaluating the value or gradient of the target distribution. Then, the target can be optimized via stochastic gradient descent. It is desirable to reduce the variance of the gradient estimate, since this governs convergence. Control variates (CVs), a classical technique from statistics, is often used to accomplish this.

This paper investigates how to use many CVs in concert. We present a systematic view of existing CVs, which starts by splitting the exact gradient into four terms (Eq. 2). Then, a CV is obtained by application of a generic "recipe": Pick a term, possibly approximate it, and take the difference of two estimators (Fig. 2). This suggests many possible CVs, including some seemingly not used before.

With many possible CVs, one can naturally ask how to use many together. In principle, the optimal combination is well known (Eq. 6). However, this requires unknown (intractable) expectations. We address this using decision theory. The goal is a "decision rule" that takes a minibatch of evaluations together with the set of CVs to be used, and returns a gradient estimate. We adopt a Bayesian risk measuring how gradient variance impacts convergence rates of stochastic optimization, with simple prior over gradients and sets of CVs. A simple optimal decision rule emerges, where the intractable expectations are replaced with "regularized" empirical estimates (Thm 4.1). To share information across iterations, we suggest combining this Bayesian approach with exponential averaging by using an "effective" minibatch size.

We demonstrate practicality on logistic regression problems, where careful combination of many CVs improves performance. For *all* learning rates, convergence is improved over any single CV.

## 1.1 Contributions

The contribution of this work is twofold. First, in Section 3, we propose a systematic view of how to generate many existing control variates. Second, we propose a an algorithm to use multiple control variates simultaneously, described in Section 4. As shown in Section 5, combining these two ideas result in gradients with low variance that allow the use of larger learning rates, while retaining convergence.

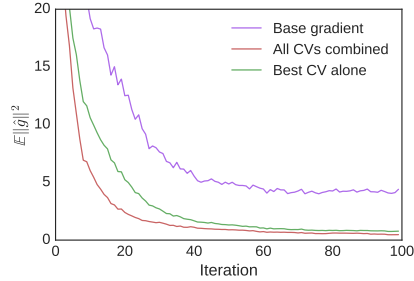

Figure 1: An example of how combining control variates reduces gradient variance for the same sequence of weights (australian dataset).

## 2 Preliminaries

Variational Inference (VI) works by transforming an inference problem into an optimization, by decomposing the marginal likelihood of the observed data $x$ given latent variables $z$ as:

$$\log p(x) = \underbrace{\mathbb{E}_{Z \sim q_w(Z)} \left[ \log \frac{p(Z,x)}{q_w(Z)} \right]}_{\text{ELBO}(w)} + \underbrace{\text{KL}(q_w(Z)||p(Z|x))}_{\text{KL-divergence}}.$$

Here, the variational distribution $q_w(z)$ is used to approximate the true posterior distribution $p(z|x)$. VI's goal is to find the parameters $w$ that minimize the KL-divergence between $q_w(z)$ and the true posterior $p(z|x)$. Since $\log p(x)$ does not depend on $w$, minimizing the KL-divergence is equivalent to maximizing the ELBO (Evidence Lower BOund).

Historically, models and variational families for which expectations were simple enough to allow closed-form updates of $w$ were used [2, 3, 32]. However, for more complex models, closed form expressions are usually not available, which has led to widespread use of stochastic optimization methods [8, 18, 19, 20, 26]. These require approximating the target's gradient

$$g(w) = \nabla_w \text{ELBO}(w) = \nabla_w \mathbb{E}_{Z \sim q_w(Z)} \left[ \log p(Z,x) - \log q_w(Z) \right]. \tag{1}$$

Good gradient estimates play an important role, since high variance will negatively impact on convergence and optimization speed. Several methods have been developed to improve gradient estimates, including Rao-Blackwellization [20], control variates [7, 17, 18, 19, 20, 28, 30, 33], closed-form solutions for certain expectations [27], discarding terms [23], and different estimators.

### 2.1 Control variates

A control variate (CV) is a random variable with expectation zero that is added to another random variable in the hope of reducing variance. Let $X$ be a random variable with unknown mean, and let $C$ be a random variable with mean zero. Then for any scalar $a$, $Y = X + a\,C$ has the same expectation as $X$ but (usually) different variance. A standard result from statistics is that the value of $a$ that minimizes the variance of $Y$ is $a = \text{Cov}(X,C)/\text{Var}(C)$, for which $\text{Var}(Y) = \text{Var}(X)(1 - \text{Corr}(X,C)^2)$. Thus, a good control variate for $X$ is a random variable $C$ that is *highly correlated* with $X$.

## 3 Systematic generation of control variates

This section gives a generic recipe for creating control variates (Fig. 2) and reviews how existing control variates are an instance of it (see also Sec. 6.4 in the appendix). We begin by splitting the ELBO gradient into four terms as

$$g(w) = \underbrace{\nabla_w \mathbb{E}_{q_w} \log p(x|Z)}_{g_1(w): \text{ Data term}} + \underbrace{\nabla_w \mathbb{E}_{q_w} \log p(Z)}_{g_2(w): \text{ Prior term}} - \underbrace{\nabla_w \mathbb{E}_{q_w} \log q_v(Z)\big|_{v=w}}_{g_3(w): \text{ Variational term}} - \underbrace{\nabla_w \mathbb{E}_{q_v} \log q_w(Z)\big|_{v=w}}_{g_4(w): \text{ Score term}}.$$

$$\tag{2}$$

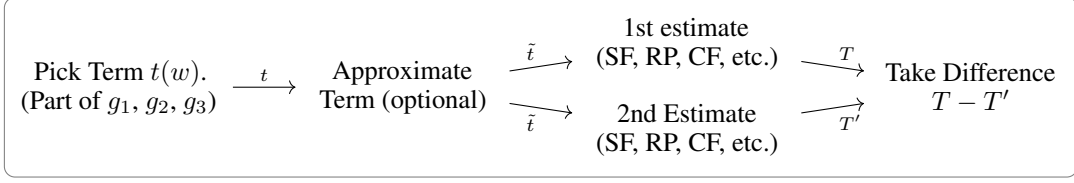

Figure 2: Generic control variate recipe. (SF: score function RP: reparameterization CF: closed form.) Sec. 6.4 (appendix) casts several existing ideas [23, 19, 17, 30, 28, 7] as instances of this recipe.

The first three terms all correspond to the influence of $w$ on the expectation of some function independent of $w$. Control variates for these terms, and for any combination of them, are discussed in Sec. 3.1-3.2. The score term, discussed in Sec. 3.3, is different, since the function *inside* the expectation depends on $w$. (Roeder et al. [23] give a related decomposition, albeit specifically for reparameterization estimators.)

### 3.1 Control Variates from Pairs of estimators

The basic technique for deriving CVs is to take the *difference* between a pair of unbiased estimators of a general term $t(w)$ (any of $g_1$, $g_2$, $g_3$ or a combination of them), which must therefore have expectation zero. The terms $g_1$, $g_2$ or $g_3$ are all the expectation (over $q_w$) of some function $f$ (independent of $w$). [1] Thus, $t(w)$ can be written as

$$t(w) = \nabla_w \mathop{\mathbb{E}}_{q_w(Z)} [f(Z)] \qquad \text{or} \qquad \left( \nabla_w \mathop{\mathbb{E}}_{q_w(Z)} [f_v(Z)] \right) \Big|_{v=w}.$$

Many methods exist to estimate gradients of this type. Mathematically, we think of these as *random variables* (with a corresponding generation algorithm). A few estimators are summarized in Eq. 3 (dropping dependence of $f$ on $v$). If we write $T^a$ for an estimator for $t(w)$ using method $a$, then

$$t(w) = \mathbb{E} \begin{cases} T^{SF} &= f(Z) \nabla_w \log q_w(Z) & \text{Score function} & Z \sim q_w \\[2mm] T^{RP_1} &= \nabla_w f(\mathcal{T}_w^1(\epsilon)) & \text{Reparameterization} & \epsilon \sim \bar{q} \\[2mm] T^{RP_2} &= \nabla_w f(\mathcal{T}_w^2(\epsilon)) & \text{Other Reparam.} & \epsilon \sim \bar{q} \\[2mm] T^{GR} &= f(Z) \nabla_w \log q_w(Z) + \nabla_w f(\mathcal{T}_w(\epsilon)) & \text{Gen. Reparam.} & \epsilon \sim \bar{q}_w, Z = \mathcal{T}_w(\epsilon) \\[2mm] T^{CF} &= \nabla_w \mathbb{E}_{q_w}[f(Z)] & \text{Closed Form} \end{cases}$$

$$(3)$$

**Score function (SF)** estimation, or REINFORCE [31], uses the equality $\nabla_w q_w(z) = q_w(z) \nabla_w \log q_w(z)$ [18, 20]. This gives $t(w) = \mathbb{E}_{q_w} T^{SF}$, with $T^{SF}$ as in Eq. 3. Unbiased estimates for the gradient can be obtained using Monte Carlo sampling, with samples from $q_w(z)$.

**Reparameterization (RP)** estimators [13, 17, 26] are based on splitting the procedure to sample from $q_w$ into sampling and transformation steps. First, sample $\epsilon \sim \bar{q}(\epsilon)$; second, transform $z = \mathcal{T}_w(\epsilon)$. Here, $\bar{q}$ is a fixed distribution (indep. of $w$) and $\mathcal{T}_w$ is a deterministic transformation. When sampling is done this way, it follows that $\mathbb{E}_{q_w} f(Z) = \mathbb{E}_{\bar{q}} f(\mathcal{T}_w(\epsilon))$, rendering the expectation independent of $w$. The general term can therefore be written as $t(w) = \mathbb{E}_{\bar{q}} T^{RP}$, with $T^{RP} = \nabla_w f(\mathcal{T}_w(\epsilon))$. The multivariate Gaussian distribution $\mathcal{N}(\mu_w, \Sigma_w)$ illustrates this: A sample can be generated by drawing $\epsilon \sim \mathcal{N}(0, I)$ and setting $\mathcal{T}_w(\epsilon) = M_w \epsilon + \mu_w$, where $M_w$ is a matrix such that $M_w M_w^T = \Sigma_w$.

**Multiple reparameterizations** are typically possible. For example, the above estimator for the multivariate Gaussian is valid with *any* $M_w$ such that $M_w M_w^T = \Sigma_w$. For instance, $M_w$ could be a lower triangular matrix obtained via the Cholesky factorization of $\Sigma_w$ [4, 26]. (Often, entries of $w$ directly specify entries in the Cholesky factorization, obviating the need to explicitly compute it.) Another option is the matrix square root of $\Sigma_w$ [14]. All valid reparameterizations give unbiased gradients, but with different statistical properties.

**Generalized reparameterization (GR)** is intended for distributions where reparameterization is not applicable, e.g. the gamma or beta [24]. Take a transformation $\mathcal{T}_w$ and a base distribution $\bar{q}_w(\epsilon)$ (*both* dependent on $w$) such that $\mathcal{T}_w(\epsilon)$ is distributed identically to $q_w(Z)$. Then, $\mathbb{E}_{q_w} f(Z) = \mathbb{E}_{\bar{q}_w} f(\mathcal{T}_w(\epsilon))$. The dependence of this expectation on $w$ is mediated partially through $w$'s influence on $\bar{q}_w$ and partially through $w$'s influence on $\mathcal{T}_w$. This leads to a representation of a general term as $t(w) = \mathbb{E}_{\bar{q}_w(\epsilon)} T^{GR}$, where $T^{GR}$ is as in Eq. 3. This has essentially has a score function-like term and a reparameterization-like term, corresponding to $w$'s influence on $\bar{q}$ and $\mathcal{T}_w$, respectively.

**Closed form (CF)** expressions are sometimes available for general terms involving $g_2$ and $g_3$, but rarely for $g_1$. This is because a closed-form expression needs $q$ and $f$ to be simple enough, that is rarely the case for the data term $g_1$, which is usually estimated with one of the methods described above [17, 19, 20, 24]. However, there are some cases for which $g_1$ can be computed exactly [4].

**Data Subsampling** is often applied to the data term $g_1$ [12]. If the likelihood treats $x$ as i.i.d., then $f(z) = \log p(x|z)$ can be approximated without bias from a minibatch of data. If $f_d(z)$ is that estimate, an equivalent representation of the data term is $g_1(w) = \mathbb{E}_D \nabla_w \mathbb{E}_{q_w(Z)} f_D(Z)$ where $D$ is uniform over subsets of data. Thus, one can define an unbiased estimator by using one of the techniques above (to cope with $\mathbb{E}_{q_w(Z)}$) on a random minibatch $D$ (to cope with $\mathbb{E}_D$). With large datasets this can be much faster, but sampling $D$ acts as an additional source of variance.

## 3.2 Control Variates from approximations

The previous section used that the difference of two unbiased estimators of a term has expectation zero, and so is a control variate. Another class of control variates uses the insight that if a general term $t(w)$ is replaced with an *approximation*, the difference between two estimators of the (approximate) general term still produces a valid control variate. The motivation is that approximations might allow the use of high-quality estimators (e.g. a closed-form) not otherwise available.

Fundamentally, the randomness in the above estimators is due to two types of sampling. First, expectations over $q_w$ are approximated by sampling, introducing "distributional sampling error". Second, with large data, the data term can be approximated by drawing a minibatch, introducing "data subsampling error". Approximations to terms have been devised so that expectations (either over $q_w$ or the full dataset) can be efficiently computed.

**Correcting for distributional sampling:** Here, the goal is to approximate $f$ with some function $\tilde{f}$ so as to make $\mathbb{E}[\tilde{f}(Z)]$ easier to estimate – typically so admits a closed-form solution. Paisley et al. [19] approximate the data term with either a Taylor approximation in $z$ or a bound and then define a control variate as the difference between $\mathbb{E}[\tilde{f}(Z)]$ computed exactly and its estimator using the score function method, which greatly reduces the variance of their gradient estimate, obtained with the score function method. Miller et al. [17] also use a Taylor approximation of the data term, but use the difference between $\mathbb{E}[\tilde{f}(Z)]$ computed exactly and and its estimator using reparameterization. They use this control variate together with a base gradient estimate obtained via reparameterization.

**Correcting for data subsampling:** As discussed in Sec. 3.1 it is common with large datasets to define estimators for the data term that only evaluate the likelihood on random subsets of data. To reduce the variance introduced by this subsampling, Wang et al. [30] propose to approximate $f_d(z)$ with a Taylor expansion in $x$, leading to an approximate data term $\tilde{g}_1(z) = \nabla_w \mathbb{E}_{q_w} \mathbb{E}_D \tilde{f}_D(z)$. For some models the inner expectation (over $D$) can be computed efficiently by caching the $1^{st}$ and $2^{nd}$ order empirical moments of the data. Since the outer expectation (over $q_w$) usually remains intractable, a final control variate is obtained by applying one of the estimation methods described in Sec. 3.1 (SF, RP, etc) to both $f_D(z)$ and $\mathbb{E}_D f_D(z)$ and taking the difference.

Both correction mechanisms described above represent particular scenarios that are included in the proposed framework shown in Fig. 2, which also includes other control variates based on approximations. First, it imposes no restrictions on other approximations, such as the ones based on approximating the distribution $q_w$ instead of $f$. And second, it includes control variates based on the difference of two estimates of an approximate general term, despite neither being CF. These two ideas are used in the control variate introduced by Tucker et al. [28], which use a continuous relaxation [9, 16] to approximate the distribution $q_w$ (discrete in this case), and construct a control variate by taking the difference between the SF and RP estimates of the resulting term based on the relaxation.

Following a similar idea, Grathwohl, et al. [7] use a neural network as a surrogate for $f$, and use as control variate the difference between the SF and RP estimation of the term involving the surrogate.

### 3.3 Control variate from the score term ($g_4$)

It's easy to show that the score term is always zero, i.e. $g_4(w) = 0$ (proof in appendix). Thus, it does not need to be estimated. However, since it has expectation zero, one can use the naive control variate $T_4 = \nabla_w \log q_w(Z), Z \sim q_w$ [20, 23].

## 4 Combining multiple control variates

In order to use control variates we need to define a base gradient estimator $h(w) \in \mathbb{R}^D$ and a set of control variates, $\{c_1, ..., c_L\}, c_i \in \mathbb{R}^D$, that we want to use to reduce the base gradient's variance. We multiply each control variate $c_i$ with a scalar weight $a_i$ to get the estimator

$$\hat{g}(w) = h(w) + \sum_{i=1}^{L} a_i \, c_i(w). \tag{4}$$

Defining $a \in \mathbb{R}^L$ as the vector of weights and $C \in \mathbb{R}^{D \times L}$ as the matrix with $c_i$ as the i-th column, $\hat{g}$ can be equivalently expressed as

$$\hat{g}(w) = h(w) + C(w)a. \tag{5}$$

The goal is to find $a$ such that the final gradient has low variance. This follows from theoretical results on stochastic optimization with a first-order unbiased gradient oracle that indicate that convergence is governed by the expected squared norm $\mathbb{E}\|\hat{g}\|^2$ of the gradient oracle [1], which is equivalent (up to a constant) to the trace of the variance. In particular, in the case in which the CVs are all differences between unbiased estimators for different terms, finding the optimal $a$ is equivalent to finding the best affine combination of the estimators.[2]

**Lemma 4.1.** *Let $h(w) \in \mathbb{R}^D$ be a random variable, $C(w) \in \mathbb{R}^{L \times D}$ a matrix of random variables such that each element has mean zero. For $a \in \mathbb{R}^L$, define $\hat{g}(w) = h(w) + C(w)a$. The value of $a$ that minimizes $\mathbb{E}\|\hat{g}(w)\|^2$ for a given $w$ is*

$$a^*(w) = - \underset{p(C,h|w)}{\mathbb{E}} \left[C^T C\right]^{-1} \mathbb{E} \left[C^T h\right]. \tag{6}$$

Variants of this result are known [30]. Of course, this requires the expectations $\mathbb{E}[C^T C]$ and $\mathbb{E}[C^T h]$, which are usually not available in closed form. One solution is, given some observed gradients $h_1, ..., h_M$ and control variates $C_1, ..., C_M$, to estimate $a^*$ using empirical expectations in place of the true ones. However, this approach does not account for how errors in the estimates of these expectations affect $a$ and therefore the final variance of $\hat{g}$.

### 4.1 Bayesian regularization

We deal with this problem from a "risk minimization" perspective. We imagine that the joint distribution over $C$ and $h$ is governed by some (unknown) parameter vector $\theta$. Then, we can define the loss for selecting the vector of weights $a$ when the true parameter vector is $\theta$ as

$$L(a, \theta) = \underset{C,h|\theta}{\mathbb{E}} \|h + Ca\|^2.$$

We seek a "decision rule"

$$\alpha(C_1, h_1, ..., C_M, h_M)$$

that takes as input a "minibatch" of $M$ evaluations of $h$ and $C$ and returns a weight vector $a$. Then, for a pre-specified probabilistic model $p(C, h, \theta)$, we can define the Bayesian regret as

$$\text{BayesRegret}(\alpha) = \underset{\theta}{\mathbb{E}} \ \underset{C_1, h_1, ..., C_M, h_M | \theta}{\mathbb{E}} \left[ L \left( \alpha(C_1, h_1, ..., C_M, h_M), \theta \right) \right].$$

The following theorem shows that if we model $p(C, h | \theta)$ jointly as a Gaussian with canonical parameters $\theta = (\eta, \Lambda)$, and use a Normal-Wishart prior for $p(\theta)$, then the decision rule $\alpha$ minimizing the Bayesian risk ends up being similar to Eq. 6, with two modifications. First, the unknown expectations are replaced with empirical expectations. Second, the empirical expectation of $C^T C$ is "regularized" by a term determined by the prior. For simplicity, the following result is stated assuming that the Normal-Wishart prior uses $V_0$ being a constant times the identity. However, in the appendix we state (and prove) a more general result where $V_0$ is arbitrary. This can also be implemented efficiently, although the result is more clumsy to state.

**Theorem 4.1.** *If $p(C, h | \theta)$ is a Gaussian parameterized as*

$$p(C, h | \theta = (\eta, \Lambda)) = \text{Gaussian}\left( \left[ \text{vec}(C), h \right] \Big| \mu = \Lambda^{-1} \eta, \Sigma = \Lambda^{-1} \right),$$

*and the prior is a Normal-Wishart, parameterized as $p(\theta = (\eta, \Lambda)) \propto \exp(t_0^T \eta - \text{trace}(V_0^T \Lambda) - n_0 A(\eta, \Lambda))$, then the decision rule that minimizes the Bayesian regret for $V_0 = v_0 I$ is*

$$\alpha^*(C_1, h_1, ..., C_M, h_M) = -\left( \frac{d \, v_0}{M} I + \overline{C^T C} \right)^{-1} \overline{C^T h} \tag{7}$$

*Where $h \in \mathbb{R}^d$, $\overline{C^T C} = \frac{1}{M} \sum_{m=1}^M C_m C_m^T$ and $\overline{C^T h} = \frac{1}{M} \sum_{m=1}^M C_m^T h_m$.*

The proof idea is as follows: Since the loss is the expected squared norm, the optimal decision rule can be reduced to a form similar to Eq. 6 but with the expectations replaced by *posterior expectations* conditioned on the observations $C_1, ..., C_M$ and $h_1, ..., h_M$. For exponential families with conjugate priors (e.g. the Gaussian with a Normal-Wishart prior), the posterior expectation of sufficient statistics given observations has a simple closed-form solution [10]. The sufficient statistics for the Gaussian are the first and second joint moments of $[\text{vec}(C), h]$, from which the expectations needed for the optimal decision rule can be extracted.

The rule in Eq. 7 is surprisingly simple: just compute the empirical averages and add a diagonal regularizer before solving the linear system. Using a large $M$ provides better estimates for the expectation and thus reduces the amount of "regularization" applied, while using a small $M$ provides worse estimates, which are regularized more heavily.

### 4.2 Empirical Averages

The probabilistic model described above does not explicitly mention the parameters $w$. One way to use this would be to apply it separately in each iteration. It is desirable, however, to exploit the fact that the parameters change slowly during learning. Algorithmically, the procedure above requires as input only empirical expectations for $C^T C$ and $C^T h$. Instead of using samples from a single step alone, we propose using an exponential average. At every step we compute a weighted average of the previous empirical expectation and the current one. This results in the update rule $\overline{E}_t = (1 - \gamma)\overline{E}_{t-1} + \gamma \hat{E}_t, \, ; \, \gamma \in [0, 1]$ where $E$ represents either $C^T C$ or $C^T h$, and $\hat{E}_t$ is the empirical average obtained using the samples drawn at step $t$. To combine this with the Bayesian regularization procedure, we use an "effective M", $M_{eff} = B \sum_{t=1}^T (1 - \gamma)^t$, which indicates how many samples are effectively being included in the empirical averages, where $B$ is the minibatch size. $M_{eff}$ is used instead of $M$ in equation 7. Technically, the regularization procedure assumes that the samples for the empirical expectations are independent of those actually used for the final gradient estimate $\hat{g}$. To reflect this, we compute $\alpha$ at step $t$ using the empirical average from step $t - 1$, $\overline{E}_{t-1}$.

## 5 Experiments and Results

We tried several control variates and the combination algorithm on a Bayesian binary logistic regression model with a standard Gaussian prior, using three well known datasets: ionosphere, australian, and sonar. We use simple SGD with momentum ($\beta = 0.9$) as our optimization algorithm,

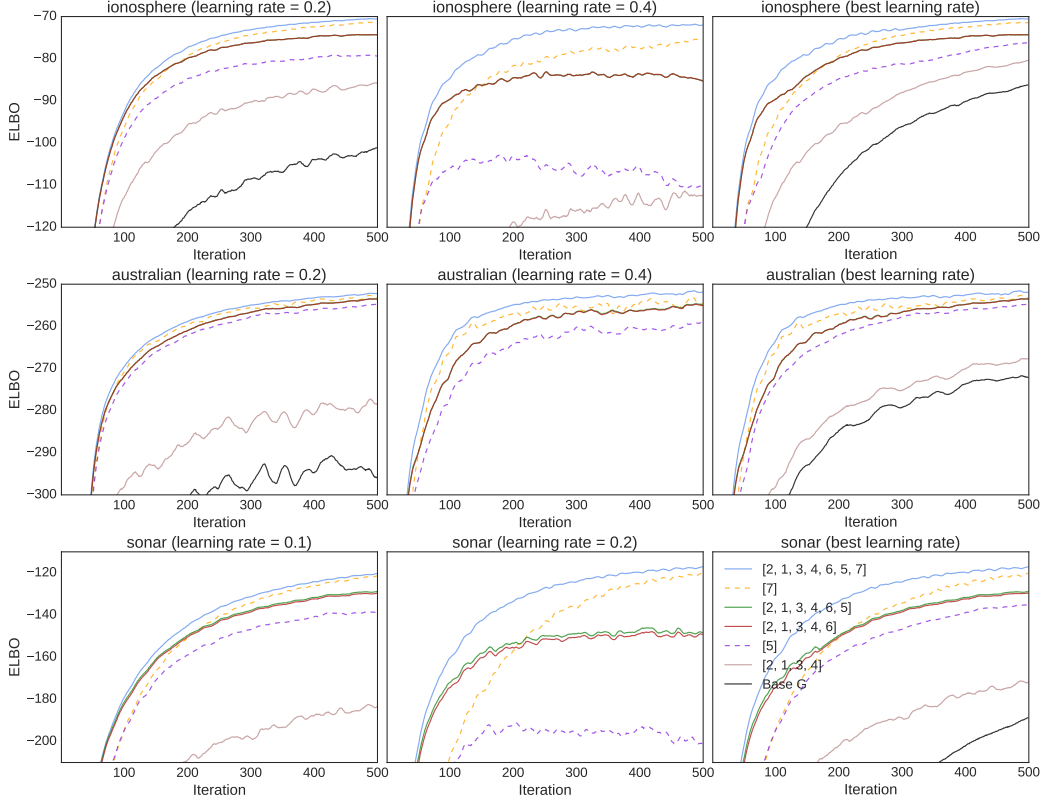

Figure 3: For each dataset, optimization results for different gradients with different learning rates. Legends indicate what control variates are used together with the base gradient. The right column shows results with the best learning rate retrospectively selected for each iteration. For clarity we limit the y-axis of the plots, which leaves some of the results (worst ones) out of the range being plot.

minibatches of size 10, a decay factor of $\gamma = 0.02$ for the exponentially decayed empirical averages, and $v_0 = 10^{-3}$, value based on results obtained for the sensitivity analysis carried out (see Sec. 5.1). We chose a full covariance Gaussian as variational distribution $q_w(z)$ parameterized using the mean and a Cholesky factorization of the covariance. Since both the prior and the variational distribution are Gaussian, the prior and variational terms can be computed in closed form.

As base gradient we use what seems to be the most common estimator, with reparameterization ($RP_1$) to estimate the data term $g_1$ (with the local reparameterization trick [12]) and the prior term $g_2$, and a closed form expression for the variational/entropy term $g_3$. Here, $RP_1$ is the reparameterization estimator using $\mathcal{T}(\epsilon; w) = \text{Cholesky}(\Sigma_w)\epsilon + \mu_w$, while $RP_2$ uses $\mathcal{T}(\epsilon; w) = \sqrt{\Sigma_w}\epsilon + \mu_w$ [14] with the matrix square root. For CVs, we chose to use the following seven, which provide a reasonable coverage of the different methods described in Section 3:

- $c_1$: The difference between the $RP_1$ and closed-form estimates of the variational term.
- $c_2$: The difference between the $RP_1$ and closed-form estimates of the prior term.
- $c_3$: The difference between the $RP_1$ and $RP_2$ estimates of the prior term.
- $c_4$: The difference between the $RP_1$ and $RP_2$ estimates of the data term.
- $c_5$: Taylor expansion of the $RP_1$ estimate of the data term, correcting for data subsampling [30].
- $c_6$: Taylor expansion of the $RP_2$ estimate of the data term, correcting for data subsampling [30].
- $c_7$: Taylor expansion of the $RP_1$ estimate of the data term, correcting for sampling from $q_w(z)$. This control variate is based on the work of Miller, et al [17], but adapted to a full covariance (rather than diagonal) Gaussian (see appendix).

We compare the optimization results obtained using the base gradient alone and the base gradient combined with different subsets of CVs, which were chosen following a simple approach: We tried each

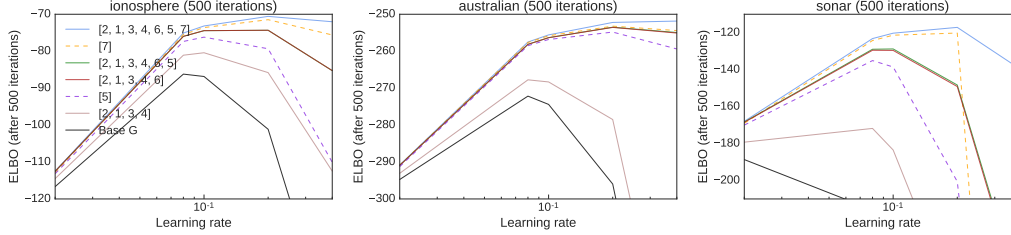

Figure 4: ELBO after 500 iterations for each gradient vs. learning rate (legends as in Fig. 3).

CV in isolation, and chose the four worst performing ones as one subset, the five worst performing ones as another subset, and so on. The final subsets of CVs obtained this way are $S_4 = \{c_2, c_1, c_3, c_4\}$, $S_5 = \{c_2, c_1, c_3, c_4, c_6\}$, $S_6 = \{c_2, c_1, c_3, c_4, c_6, c_5\}$, and $S_7 = \{c_2, c_1, c_3, c_4, c_6, c_5, c_7\}$. We also show results for the two best control variates, $c_5$ and $c_7$, used in isolation. All the results shown in this section, figures and tables, were obtained averaging the results from 50 runs.

**Best learning rate.** Table 1 shows the ELBO value achieved after 500 iterations, with the largest learning rate[3] for which optimization converged with at least one estimator. It can be seen that increasing the number of CVs often leads to higher final values for the ELBO and that, in all cases, the higher ELBOs (better) were achieved by using all CVs together.

Table 1: Average ELBO achieved after 500 iterations for each dataset using the base gradient with different subsets of control variates and particular learning rates (lr).

| Dataset (lr) | Control variates used | | | | | | |
|---|---|---|---|---|---|---|---|
| | - | $S_4$ | $S_5$ | $S_6$ | $S_7$ | $c_5$ | $c_7$ |
| Ion. (0.4) | $-157.3$ | $-112.5$ | $-85.3$ | $-85.3$ | $-72$ | $-110.1$ | $-75.6$ |
| Aus. (0.4) | $-378.2$ | $-357.2$ | $-255.1$ | $-255$ | $-251.8$ | $-259.4$ | $-254.4$ |
| Sonar (0.2) | $-442.4$ | $-270.2$ | $-149.1$ | $-148.3$ | $-117.1$ | $-200.6$ | $-120.2$ |

**Comparing across learning rates.** Now we compare the performance achieved using each gradient estimator with different learning rates. To do so we present two sets of images. First, the two leftmost columns of Fig. 3 show, for each dataset, the ELBO vs. iterations for two different learning rates; while the third column shows, for each gradient estimator and iteration, the ELBO for the best learning rate (vs. iteration). As in Table 1 it can be seen that for a given learning rate (or when choosing the best at each iteration) the gradients that combine more control variates are better suited for optimization and display a strictly dominant performance.

Finally, Fig. 4 shows, for several gradients, the final ELBO (after 500 iterations) vs. learning rate used, providing a systematic comparison of how the gradient estimates perform with different learning rates. Again, estimates employing more CVs display a dominant performance, with larger improvements at larger learning rates. Furthermore, the "best" learning rate increases with better estimators.

## 5.1 Sensitivity analysis

It is natural to ask how the variance of the gradient estimate is related to the choice of the prior parameter $v_0$ and the minibatch size $M$. Recall from Thm. 4.1 that a larger value of $v_0$ corresponds to a more concentrated prior, and is thus a more conservative choice – essentially it results in more "regularization" of the empirical moments. To answer this we carried out a simple experiment, where we fix $w$ and estimate $\mathbb{E} \, ||\hat{g}(w)||^2$ with a variety of $v_0$ and $M$. To choose $w$, we applied SGD with a low-variance gradient (computed with many samples), and a learning rate of $0.08$ and same initialization as in the previous section, and selected the parameters found after 25 iterations. This is intended to be "typical", in that it is neither at the start nor the end of optimization.

Estimating $\hat{g}(w)$ is a three step process: (1) Use one set of evaluations of $C$ and $h$ to estimate $E[C^T C]$ and $E[C^T h]$. (2) Apply the prior to compute $a$ from those estimate (Eq. 7). Recall from

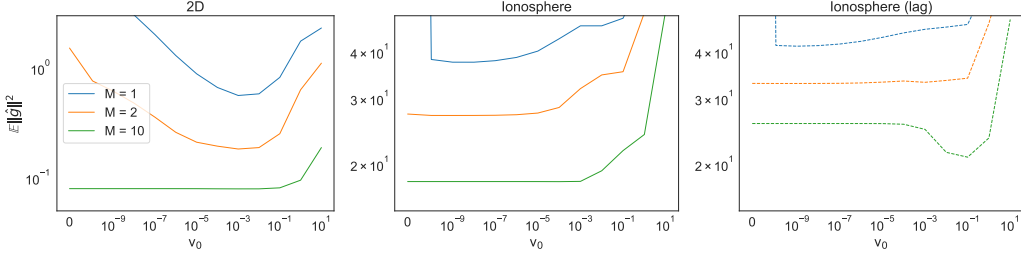

Figure 5: Expected squared norm of the gradient estimate vs $v_0$, for different minibatch sizes. The two left most images were obtained with estimates of $E[C^T C]$ and $E[C^T h]$ using the current weights $w$, while for the image on the right moments were estimated using gradients from an older iteration. The sonar and australian datasets give results similar to those of ionosphere.

Thm. 4.1 that a larger $v_0$ corresponds to a more concentrated prior, essentially "regularizing" more. (3) Use a second set of evaluations of $C$ and $h$ to compute $\hat{g}(w)$, using weights $a$ (Eq. 4 / 5).

In a first experiment, we tested exactly that procedure, drawing two independent evaluations of $C$ and $h$ using the current weights $w$. Results are shown in Figure 5. We found a small artificial dataset illustrative, with samples $x \in \mathbb{R}^2$. For this "2D" dataset, with small minibatches, a fairly large value of $v_0$ provided the best results. However with ionosphere, even a very small $v_0$ tended to perform well.

For efficiency, our logistic regression experiments used exponential averaging over previous iterations to estimate $E[C^T C]$ and $E[C^T h]$, rather than drawing two evaluations at each iteration. So, even with large value of $M$ these are not fully reliable. To roughly simulate this, we performed a second "lagged" experiment estimating $E[C^T C]$ and $E[C^T h]$ from evaluations of $C$ and $h$ at the weight from 10 iterations previous during SGD. (This was chosen considering the "average age" of gradients when using exponential averaging, and that $0.08$ is a relatively small learning rate.) The results of this are shown on the right of Fig. 5. Lagged evaluations result in stochastic gradients with more variance, with a different dependence on $v_0$. (Note, however, that the gradient remains unbiased, lagging is cheaper, and that all estimators have a variance decreasing with $M$.)

We emphasize that several somewhat arbitrary decisions were made for these experiments, such as the learning rate, the choice of iteration, the amount of "lag". However, we believe that the results illustrate an important phenomenon related to the use of regularization: when using past gradient information (as exponential averaging does) larger values of $v_0$ are beneficial and result in gradients with lower variance. While intuitively plausible, note that this benefit of regularization for countering errors introduced by the use of old gradients is not really captured by our theoretical analysis in Section 4 which is entirely based on "single-iteration" reasoning.

## 6    Conclusion

This work focuses on how to obtain low variance gradients given a fixed set of control variates. We first present a unified view that attempts to explain how most control variates used for variational inference are derived, which sheds light on the large number of CVs available. We then propose a combination algorithm to use multiple control variates in concert. We show experimentally that, given a set of control variates, the combination algorithm provides a simple and effective combination rule that leads to gradients with less variance than those obtained using a reduced number of CVs (or no CVs at all). The algorithm assumes that a fixed set of control variates to be used is given, and minimizes the final gradient's variance using them, without analyzing how favorable using all the CVs actually is. A "smarter" algorithm could, for instance, decide whether to use all the CVs given or a just a subset. We leave the development of such algorithm for future work.

## Footnotes

[1] For $g_1$, $g_2$, and $g_3$, use $f(z) = \log p(x|z)$, $f(z) = \log p(z)$, and $f_v(z) = \log q_v(z)$ respectively.

[2]Intuitively, given two estimators, if one is used as the base estimator and the difference as a CV, then finding the best weight for that CV is equivalent to finding the best mixture of the estimators.

[3]The loss is normalized by the number of samples in the dataset. If it was not the equivalent learning rates would be smaller

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
