[Supplementary Material]

# Appendix

## 6.1 Proof of Lemmas

**Lemma 6.1.** $E_{q_w(z)}\big[\nabla_w \log q_w(Z)\big] = 0$

*Proof.*

$$E_{q_w(z)}\big[\nabla_w \log q_w(Z)\big] = E_{q_w(z)}\left[\frac{\nabla_w q_w(Z)}{q_w(Z)}\right] = \int q_w(z)\frac{\nabla_w q_w(z)}{q_w(z)}\mathrm{dz} = \nabla_w \int q_w(z)\mathrm{dz} = \nabla_w 1 = 0$$

$\square$

**Lemma 6.2.** *Let $h(w) \in \mathbb{R}^D$ be a random variable, $C(w) \in \mathbb{R}^{L \times D}$ a matrix of random variables such that each element has mean zero. For $a \in \mathbb{R}^L$, define $\hat{g}(w) = h(w) + C(w)a$. The value of $a$ that minimizes $\mathbb{E}\left\|\hat{g}(w)\right\|^2$ for a given $w$ is*

$$a^*(w) = - \underset{p(C,h|w)}{\mathbb{E}} \big[C^T C\big]^{-1} \mathbb{E}\big[C^T h\big]. \tag{6}$$

*Proof.*

$$
\begin{aligned}
E[\hat{g}^T \hat{g}] &= E[(h + Ca)^T (h + Ca)] \\
&= E[h^T h + 2h^T Ca + a^T C^T Ca] \\
&= E[h^T h] + 2E[h^T C]a + a^T E[C^T C]a
\end{aligned}
$$

Differentiating with respect to $a$, and making the result equal to 0 gives us

$$2E[h^T C] + 2E[C^T C]a = 0 \longrightarrow a^* = -E[C^T C]^{-1}E[C^T h]$$

$\square$

## 6.2 Proof of Theorem 4.1

First we state Lemma 6.3 and Lemma 6.4, which will use to prove the main theorem 4.1.

**Lemma 6.3.** *Suppose that*

$$P(x|w) = h(x)\exp(\langle w, T(x)\rangle - A(w))$$

*be some exponential family, and let*

$$P(w|\tau_0) \propto \exp\left(\langle \tau_0, w\rangle - n_0 A(w)\right)$$

*be the conjugate prior to that family. If $x_1, ..., x_N$ are i.i.d. variables and $X$ is new data from the distribution, then*

$$E[T(X)|x_1, ..., x_N] = \kappa \frac{\tau_0}{n_0} + (1 - \kappa)\hat{\mu},$$

*where $\hat{\mu} = \frac{1}{N}\sum_{n=1}^{N} T(x_n)$ and $\kappa = \frac{n_0}{n_0 + N}$.*

For a proof see Jordan [10].

**Lemma 6.4.** *Given some observations $C_1, h_1, ..., C_M, h_M$, the decision rule that minimizes the Bayes regret is*

$$a(C_1, h_1, ..., C_M, h_M) = \mathbb{E}[CC^T|C_1, h_1, ..., C_M, h_M]^{-1} \mathbb{E}[Ch|C_1, h_1, ..., C_M, h_M]$$

*Where the expectations are over all possible values of $\theta$, $C$ and $h$, given the observed data.*

*Proof.* We have

$$a^*(C_1, h_1, ..., C_M, h_M) = \text{argmin}_a \, \mathbb{E}\left[\|h + Ca\|^2 | C_1, h_1, ..., C_M, h_M\right]$$

$$= \text{argmin}_a \, \mathbb{E}\left[(h + Ca)^T(h + Ca)|C_1, h_1, ..., C_M, h_M\right]$$

The solution to this is to find the derivative of the above expression with respect to $a$ and find $a$ such that

$$0 = \mathbb{E}\left[h^T Ca + h + a^T C^T Ca | C_1, h_1, ..., C_M, h_M\right]$$

This gives $a^* = \mathbb{E}[CC^T | C_1, h_1, ..., C_M, h_M]^{-1} \, \mathbb{E}[C\,h | C_1, h_1, ..., C_M, h_M]$

$\square$

Now we prove Theorem 4.1 for an arbitrary $V_0$, using Lemma 6.3 and Lemma 6.4. Consider the same setting as in Section 4. Let $h_1, ..., h_M$ be observed gradients and $C_1, ..., C_M$ be observed control variates. We define a probabilistic model composed by the likelihood $p(h, C|\theta)$ and the prior $p(\theta)$.

**Theorem 6.5.** *If we choose the likelihood to be a Gaussian,*

$$P(h, C|\theta = (\eta, \Lambda)) = \text{Gaussian}\left(\begin{bmatrix} h \\ vec(C) \end{bmatrix} | \mu = \Lambda^{-1}\eta, \Sigma = \Lambda^{-1}\right)$$

*the prior (conjugate) to be*

$$P(\theta = (\eta, \Lambda)) \propto \exp(t_0^T \eta - tr(V_0^T \Lambda) - n_0 A(\eta, \Lambda))$$

*where $V_0$ can be written as:*

$$V_0 = \begin{bmatrix} V_{hh} & V_{hc_1}^\top & V_{hc_2}^\top & \cdots & V_{hc_L}^\top \\ V_{hc_1} & V_{c_1 c_1} & V_{c_2 c_1}^\top & \cdots & V_{c_L c_1}^\top \\ V_{hc_2} & V_{c_2 c_1} & & & \\ \vdots & & & & \vdots \\ V_{hc_L} & V_{c_L c_1} & & \cdots & V_{c_L c_L} \end{bmatrix}$$

*then the decision rule that minimizes the Bayesian regret is*

$$a^*(h_1, C_1, ..., h_M, C_M) = -E[C^T C|h_1, C_1, ..., h_M, C_M]^{-1} E[C^T h|h_1, C_1, ..., h_M, C_M] \quad (8)$$

*Where*

$$E[C^T h|h_1, C_1, ..., h_M, C_M] = \frac{\kappa}{n_0} \begin{bmatrix} trV_{hc_1} \\ trV_{hc_2} \\ \vdots \\ trV_{hc_L} \end{bmatrix} + (1 - \kappa)\overline{C^T h}.$$

*and*

$$E[C^T C|h_1, C_1, ..., h_M, C_M] = \frac{\kappa}{n_0} \begin{bmatrix} trV_{c_1 c_1} & trV_{c_2 c_1}^T & \cdots & trV_{c_L c_1}^T \\ trV_{c_2 c_1} & & & \\ \vdots & & & \vdots \\ trV_{c_L c_1} & & \cdots & trV_{c_L c_L} \end{bmatrix} + (1 - \kappa)\overline{C^T C}.$$

*With $\overline{C^T C} = \frac{1}{M}\sum_{m=1}^{M} C_m C_M^T$ being an empircal average and $\overline{C^T h} = \frac{1}{M}\sum_{m=1}^{M} C_m^T h_m$ also being an empirical average.*

*Proof.* The expression for $a^*$ in equation 8 is obtained using Lemma 6.4. We need to find $E[C^T C | h_1, C_1, ..., h_M, C_M]$ and $E[C^T h | h_1, C_1, ..., h_M, C_M]$. Using Lemma 6.3, given observations $h_1, C_1, ..., h_M, C_M$ we have a closed form expression form

$$E\left[ \begin{bmatrix} h \\ \text{vec}(C) \end{bmatrix} \begin{bmatrix} h \\ \text{vec}(C) \end{bmatrix}^T \Big| h_1, C_1, ..., h_M, C_M \right] = \kappa \frac{V_0}{n_0} + (1-\kappa) \overline{\begin{bmatrix} h \\ \text{vec}(C) \end{bmatrix} \begin{bmatrix} h \\ \text{vec}(C) \end{bmatrix}^T} \quad (9)$$

Where $\text{vec}(C)$ is a vector with the columns of $C$ concatenated. Noticing that

$$C^T h = \begin{bmatrix} c_1^T h \\ c_2^T h \\ \vdots \\ c_L^T h \end{bmatrix}$$

it can be concluded that each component of $E[C^T h | h_1, C_1, ..., h_M, C_M]$ corresponds to the sum of $d$ components of the matrix on the right hand side of equation 9. Using the decomposition for $V_0$ shown above, we get:

$$E\left[ C^T h | h_1, C_1, ..., h_M, C_M \right] = \begin{bmatrix} \text{tr} V_{hc_1} \\ \text{tr} V_{hc_2} \\ \vdots \\ \text{tr} V_{hc_L} \end{bmatrix} + (1-\kappa) \overline{C^T h}. \quad (10)$$

A similar reasoning can be used for $E[C^T C | h_1, C_1, ..., h_M, C_M]$, getting:

$$E[C^T C | h_1, C_1, ..., h_M, C_M] = \begin{bmatrix} \text{tr} V_{c_1 c_1} & \text{tr} V_{c_2 c_1}^T & \cdots & \text{tr} V_{c_L c_1}^T \\ \text{tr} V_{c_2 c_1} & & & \\ & & & \vdots \\ \text{tr} V_{c_L c_1} & & \cdots & \text{tr} V_{c_L c_L} \end{bmatrix} + (1-\kappa) \overline{C^T C} \quad (11)$$

Replacing these expressions in $a^*(h_1, C_1, ..., h_M, C_M)$ concludes the proof.

$\square$

Using the result above we now prove Theorem 4.1, which takes $V_0 = v_0 I$.

**Theorem 4.1.** *If $p(C, h | \theta)$ is a Gaussian parameterized as*

$$p(C, h | \theta = (\eta, \Lambda)) = \text{Gaussian}\left( [\text{vec}(C), h] \Big| \mu = \Lambda^{-1}\eta, \Sigma = \Lambda^{-1} \right),$$

*and the prior is a Normal-Wishart, parameterized as $p(\theta = (\eta, \Lambda)) \propto \exp(t_0^T \eta - \text{trace}(V_0^T \Lambda) - n_0 A(\eta, \Lambda))$, then the decision rule that minimizes the Bayesian regret for $V_0 = v_0 I$ is*

$$\alpha^*(C_1, h_1, ..., C_M, h_M) = -\left( \frac{d\, v_0}{M} I + \overline{C^T C} \right)^{-1} \overline{C^T h} \quad (7)$$

*Where $h \in \mathbb{R}^d$, $\overline{C^T C} = \frac{1}{M} \sum_{m=1}^{M} C_m C_m^T$ and $\overline{C^T h} = \frac{1}{M} \sum_{m=1}^{M} C_m^T h_m$.*

*Proof.* The expression for $a^*$ is given in eq. 8. We need to find the expressions for $E[C^T C | h_1, C_1, ..., h_M, C_M]$ and $E[C^T h | h_1, C_1, ..., h_M, C_M]$ when $V_0 = v_0 I$. In this particular case we get that $\text{tr} V_{hc_l} = 0$ for $l = 1, ..., L$. Combining this with eq. 10 gives

$$E[C^T h | h_1, C_1, ..., h_M, C_M] = (1-\kappa) \overline{C^T h}.$$

When $V_0 = v_0 I$ we also get

- $\mathrm{tr} V_{c_l c_k} = 0$ for $k \neq l$

- $\mathrm{tr} V_{c_l c_l} = d$

Combining these two facts with eq. 11 gives

$$E[C^T C | h_1, C_1, ..., h_M, C_M] = \frac{\kappa}{n_0} d\, v_0\, I + (1 - \kappa) \overline{C^T C}.$$

Finally,

$$
\begin{aligned}
a^*(h_1, C_1, ..., h_M, C_M) &= -\left( \tfrac{\kappa}{n_0} d\, v_0\, I + (1 - \kappa) \overline{C^T C} \right)^{-1} (1 - \kappa) \overline{C^T h} \\[2mm]
&= -\left( \tfrac{\kappa}{n_0(1-\kappa)} d\, v_0\, I + \overline{C^T C} \right)^{-1} \overline{C^T h} \\[2mm]
&= -\left( \tfrac{d\, v_0}{M} I + \overline{C^T C} \right)^{-1} \overline{C^T h}
\end{aligned}
$$

Where in the last equality we used $\kappa = \frac{n_0}{n_0 + M}$

$\square$

## 6.3 Adaptation of control variate introduced by Miller et al. [17] to full covariance Gaussians

The derivation of the variate introduced by Miller et al. [17] was done for the case in which $q_w$ is a Gaussian distribution with a diagonal covariance matrix. This section of the appendix explains how to use the CV in the case in which $q_w$ is a Gaussian distribution with full covariance matrix, in which case $w = [C_w, \mu_w]$, where $C_w C_w^T = \Sigma$ and $\mu_w$ is the mean of the distribution.

Following the procedure in Miller et al. [17], we build an approximation for $g(w) = \nabla_w \mathbb{E}_{q_w}[f(Z)]$ as $\tilde{g}(w) = \nabla_w \mathbb{E}_{q_w}[\tilde{f}(Z)] = [\nabla_{\mu_w} \mathbb{E}_{q_w} \tilde{f}(Z), \nabla_{C_w} \mathbb{E}_{q_w} \tilde{f}(Z)]$, where $\tilde{f}(z)$ is a second order Taylor expansion. The difference between $\tilde{g}(w)$ computed exactly (lemma 6.6) and its estimation using reparameterization is used as a control variate.

**Lemma 6.6.** *Let* $q_w = \mathcal{N}(\mu_w, C_w C_w^T)$ *and* $\tilde{f}(z)$ *be a second order Taylor expansion of* $f$, *then* $\nabla_{\mu_w} \mathbb{E}_{q_w} \tilde{f}(Z) = \nabla f(z_0)$ *and* $\nabla_{C_w} \mathbb{E}_{q_w} \tilde{f}(Z) = \nabla^2 f(z_0) C$.

*Proof.* Applying reparameterization we can express $\nabla_w \mathbb{E}_{q_w} \tilde{f}(Z) = \nabla_w \mathbb{E}_{\bar{q}} \tilde{f}(\mathcal{T}_w(r))$, where $r \sim \bar{q} = \mathcal{N}(0, I)$ and $\mathcal{T}_w(r) = C_w r + \mu_w$.

Introducing the Taylor expansion we get

$$
\begin{aligned}
\mathbb{E}_{\bar{q}}[\tilde{f}(\mathcal{T}_w(r))] &= \mathbb{E}_{\bar{q}}[f(z_0) + (Z - z_0)^T \nabla f(z_0) + \tfrac{1}{2}(Z - z_0)^T \nabla^2 f(z_0)(Z - z_0)]_{Z = \mathcal{T}_w(r)} \\[2mm]
&= f(z_0) + (\mathbb{E}\, \mathcal{T}_w(r) - z_0) \nabla f(z_0) + \tfrac{1}{2} \mathbb{E}\left[ \mathrm{tr}(\nabla^2 f(z_0)(\mathcal{T}_w(r) - z_0)(\mathcal{T}_w(r) - z_0)^T) \right] \\[2mm]
&= f(z_0) + (\mu_w - z_0) \nabla f(z_0) + \tfrac{1}{2} \mathrm{tr}(\nabla^2 f(z_0) \mathbb{E}[(\mathcal{T}_w(r) - z_0)(\mathcal{T}_w(r) - z_0)^T)])
\end{aligned}
$$

(12)

Where

$$\mathbb{E}[(\mathcal{T}_w(r) - z_0)(\mathcal{T}_w(r) - z_0)^T] = \mathbb{E}[(C_w r + \mu_w - z_0)(C_w r + \mu_w - z_0)^T]$$

$$= C_w \underbrace{\mathbb{E}[rr^T]}_{I} C_w^T + C_w \underbrace{\mathbb{E}[r]}_{0}(\mu_w - z_0)^T$$

$$+ (\mu_w - z_0) \underbrace{\mathbb{E}[r]}_{0} C_w^T + (\mu_w - z_0)(\mu_w - z_0)^T$$

$$= C_w C_w^T + (\mu_w - z_0)(\mu_w - z_0)^T$$

$$= C_w C_w^T + \mu_w \mu_w^T - \mu_w z_0^T - z_0 \mu_w^T + z_0 z_0^T$$

And thus

$$\mathrm{tr}(\nabla^2 f(z_0)\, \mathbb{E}[(\mathcal{T}_w(r) - z_0)(\mathcal{T}_w(r) - z_0)^T)]) = \mathrm{tr}\big(\nabla^2 f(z_0)(C_w C_w^T + \mu_w \mu_w^T - \mu_w z_0^T - z_0 \mu_w^T)\big)$$

$$= \mathrm{tr}(C_w^T \nabla^2 f(z_0) C_w)$$

$$+ \mu_w^T \nabla^2 f(z_0) \mu_w - 2 z_0^T \nabla^2 f(z_0) \mu_w \tag{13}$$

Using the results from eq. 13 in eq. 12 we get

$$\mathbb{E}_{\bar{q}}[\tilde{f}(\mathcal{T}_w(r))] = f(z_0) + (\mu_w - z_0)\nabla f(z_0)$$

$$+ \tfrac{1}{2}\mathrm{tr}(C_w^T \nabla^2 f(z_0) C_w) + \mu_w^T \nabla^2 f(z_0)\mu_w - 2z_0^T \nabla^2 f(z_0)\mu_w$$

Finally, computing the gradient $\nabla_{\mu_w} \mathbb{E}_{\bar{q}}[\nabla_w \tilde{f}(\mathcal{T}_w(r))]$ and $\nabla_{C_w} \mathbb{E}_{\bar{q}}[\nabla_w \tilde{f}(\mathcal{T}_w(r))]$ and evaluating the results in $z_0 = \mu_w$ (following [17]) yields

$$\nabla_{\mu_w} \mathbb{E}_{\bar{q}}[\tilde{f}(\mathcal{T}_w(r))]\big|_{z_0 = \mu_w} = \nabla f(\mu_w) + 2\nabla^2 f(\mu_w)\mu_w - 2\nabla^2 f(\mu_w)\mu_w$$

$$= \nabla f(\mu_w)$$

$$\nabla_{C_w} \mathbb{E}_{\bar{q}}[\tilde{f}(\mathcal{T}_w(r))]\big|_{z_0 = \mu_w} = \nabla^2 f(\mu_w) C_w$$

$$\square$$

## 6.4 Previously used Control Variates

In this section we show how many of the control variates described and used in previous work fit the proposed framework. For convenience, we repeat our generic recipe for control variates.

Also, recall our decomposition of the full gradient into different terms:

$$g(w) = \underbrace{\nabla_w \, \mathbb{E}_{q_w} \log p(x|Z)}_{g_1(w): \text{ Data term}} + \underbrace{\nabla_w \, \mathbb{E}_{q_w} \log p(Z)}_{g_2(w): \text{ Prior term}} - \underbrace{\nabla_w \, \mathbb{E}_{q_w} \log q_v(Z)\big|_{v=w}}_{g_3(w): \text{ Variational term}} - \underbrace{\nabla_w \, \mathbb{E}_{q_v} \log q_w(Z)\big|_{v=w}}_{g_4(w): \text{ Score term}}.$$

The rest of this section gives seven examples of existing control variates, and how they can be seen as instantiations of the above generic recipe.

Figure 6: Even if the exact entropy can be computed, it may be preferable to approximate it when $q_w \approx p$ [23]. This suggests a control variate consisting of the difference of the exact entropy gradient and an approximation of it. In general, it is most beneficial to include the same estimator as used to estimate gradients of the data and prior terms.

Figure 7: To estimate gradients Paisley et al. [19] approximate $f = g_1 + g_2$ using a second order Taylor expansion and upper/lower bounds, leading to $\tilde{t}(w) = \mathbb{E}_{q_w(z)} \tilde{f}(Z)$. The difference between the approximate term computed in closed form and its estimation using the score function is used as a control variate.

Figure 8: To estimate gradients Paisley et al. [19] approximate $f = g_1 + g_2$ using a second order Taylor expansion and upper/lower bounds, leading to $\tilde{t}(w) = \mathbb{E}_{q_w(z)} \tilde{f}(Z)$. The difference between the approximate term computed in closed form and its estimation using the score function is used as a control variate.

$$t = g_1 \quad \xrightarrow{\ t\ } \quad \substack{\text{Second order}\\\text{Taylor expansion}} \quad \substack{\xrightarrow{\ \tilde{t}\ }\\\xrightarrow{\ \tilde{t}\ }} \quad \substack{\text{Closed form}\\\\\text{Reparam.}} \quad \substack{\xrightarrow{\ T\ }\\\xrightarrow{\ T'\ }} \quad \substack{\text{Take Difference}\\T - T'}$$

Figure 9: To estimate gradients Miller et al. [17] approximate the data term using a second order Taylor expansion of $f$, leading to $\tilde{t}(w) = \mathbb{E}_{q_w(z)} \tilde{f}(Z)$. The difference between the approximate term computed in closed form and its estimation using reparameterization is used as a control variate.

$$t = g_1 \quad \xrightarrow{\ t\ } \quad \substack{\text{Second order}\\\text{Taylor expansion}} \quad \substack{\xrightarrow{\ \tilde{t}\ }\\\xrightarrow{\ \tilde{t}\ }} \quad \substack{\text{Reparam. } (q_w) +\\\text{Closed form } (D)\\\\\text{Reparam. } (q_w) +\\\text{Minibatch}} \quad \substack{\xrightarrow{\ T\ }\\\xrightarrow{\ T'\ }} \quad \substack{\text{Take Difference}\\T - T'}$$

Figure 10: To estimate gradients Wang et al. [30] approximate the data term using a second order Taylor expansion of $f$, for which the expectation with respect to $D$ (distribution over minibatches) can be computed in closed form. We adapt this idea to the VI setting, leading to $\tilde{t}(w) = \mathbb{E}_{q_w(z)} \mathbb{E}_D \tilde{f}_d(Z)$. The difference between the results obtained by computing the inner expectation in closed form and estimating it with a random minibatch (in both cases estimating the outer expectation using reparameterization) is used as a control variate.

$$t = g_1 + g_2 + g_3 \quad \xrightarrow{\ t\ } \quad \substack{\text{Concrete}\\\text{relaxation}} \quad \substack{\xrightarrow{\ \tilde{t}\ }\\\xrightarrow{\ \tilde{t}\ }} \quad \substack{\text{Score function}\\\\\text{Reparam.}} \quad \substack{\xrightarrow{\ T\ }\\\xrightarrow{\ T'\ }} \quad \substack{\text{Take Difference}\\T - T'}$$

Figure 11: To estimate gradients for problems with discrete variables Tucker et al. [28] use a continuous relaxation [9, 16] for the discrete variational distribution $q_w(z)$, $\tilde{q}_w(z)$, leading to $\tilde{t}(w) = \mathbb{E}_{\tilde{q}_w(z)} \tilde{f}(Z)$. Then, the difference of a score function and reparameterization estimate is used as a control variate.

$$t = g_1 + g_2 + g_3 \quad \xrightarrow{\ t\ } \quad \substack{\text{Surrogate neural}\\\text{network}} \quad \substack{\xrightarrow{\ \tilde{t}\ }\\\xrightarrow{\ \tilde{t}\ }} \quad \substack{\text{Score function}\\\\\text{Reparam.}} \quad \substack{\xrightarrow{\ T\ }\\\xrightarrow{\ T'\ }} \quad \substack{\text{Take Difference}\\T - T'}$$

Figure 12: To estimate gradients Grathwohl et al. [7] train a surrogate neural network $\tilde{f}$ to approximate $f$, leading to $\tilde{t}(w) = \mathbb{E} \tilde{f}(Z)$. Then, the difference of a score function and reparameterization estimate is used as a control variate. The neural network is trained to minimize the variance of the resulting estimator. (For discrete variational distributions they also use a continuous relaxation [9, 16] to approximate $q_w$.)