[Reviews · NeurIPS 2018]

Reviewer 1



Thank you for the thoughtful response. I have read the other reviews and the rebuttal, and after discussing the work I am electing to keep my score the same. I am somewhat unsatisfied by the author response; for papers where gradient estimator efficiency (in terms of variance) is in service of the optimization problem, comparing ELBO traces by iteration can be very misleading. If the machinery you introduce to efficiently use an ensemble of control variates is not very costly, then it should be measured or shown in your experiments. My comments below weren't about optimal tuning, they were more about exploring/understanding the sensitivity of their method on the parameters they introduce. As I understand it, the main contribution of this work is the development of a regularized estimator of what is essentially a linear regression coefficient using conjugacy in exponential families. I think that's a very clever way to approach the problem of combining control variates. But, from the experiments, I did not get a sense the sensitivity of their estimator to these prior parameters was explored. How were they set? How do they interact with the number of samples chosen to construct this estimator? If this were a model for data, we would expect some sensitivity analysis to be done to see how much prior assumptions are factoring into posterior inferences. Analogously, I would hope to see some sensitivity analysis of the proposed method. And from the rebuttal, I get the sense that these sensitivities will be left unexplored. -------------------------------- The authors present a framework for constructing control variates for stochastic optimization in variational settings. Looking at sub-parts of the variational inference objective, they define broad classes of ways to construct a control variate for each piece. They then define a way to combine multiple control variates into a single, low-variance estimator in a sample-efficient way using a regret-minimizing estimator. They compare the performance of these estimators on a logistic regression model with three different data sets at different learning rates. - How does wall clock time compare for each estimator? One issue with iteration-based comparisons is that some control variates (or their ensemble) may require more likelihood or gradient calculations. In table 1 -- how much time did each iteration take for each set of estimators? Another way to compare might be the number of likelihood evaluations required per iteration. - Figure 3 is missing a legend --- do the colors correspond to the estimators defined in the legend in Figure 2? - How does the running average parameter (gamma) from section 4.2 affect the variance of the estimator throughout optimization? This is an interesting pivot point of computational tradeoff --- fitting the optimal C with many samples at an iteration of optimization would yield the lowest variance estimator. How does that degrade when you (i) use fewer samples and rely on the InvWishart prior and (ii) use a running average of previous iterates? At a certain point it becomes not worth computing a "better C" when optimizing an objective --- where is that point? And how would you go about systematically finding it? *Quality*: I think this paper presents an interesting idea, and points to a way to use a bunch of existing estimators to get the most out of a stochastic optimization procedure. I think some of the experiments could have been conducted in a more convincing way, keeping in mind the goal of reducing the overall compute time of the optimization algorithm. I liked how the experiments section compared a wide variety of estimators, and I would like to see them probe their own contribution a little more --- how their scheme for fitting C is affected by prior assumptions, how much compute time it takes per iteration, and how the running average scheme affects optimization. *Clarity*: The paper is very clearly written and easy to follow. *Originality*: The paper compares a bunch of existing estimators within a framework that is somewhat novel. The regularization and running average of the control variate parameter is an interesting extension and a nice way to combine information from multiple estimators. However the original aspect of their paper was empirically probed the least. *Significance*: It is difficult to assess the potential impact of this method given the experiments presented. Ultimately, the utility of these gradient estimators for optimization hinge on wall clock time.

Reviewer 2



After author rebuttal: Thank you for your response. I keep my overall score and congratulate you to a very nice written paper that I very much enjoyed reading. The authors consider using ensembles of control variates rather than the typical approach of using a single one. They provide a coverage of the typical ones used in the literature and how these are constructed. The main contribution of the article is to find the optimal weights of these ensembles, and the authors take an elegant approach that minimizes a Bayesian regret and derive a closed form expression under a simplifying assumption of Gaussianity for all parameters. The value of using ensembles is clearly demonstrated in the empirical section, in which it consistently outperforms single control variates. General comments: This is a very nicely written paper which applies elegant ideas in a simple and understandable way. I find the discussion of the decomposition of the gradient of the ELBO into different terms and, moreover, how different control variates are constructed to be a perfect textbook-style review. My comments / questions: 1. Control variates reduce the variance (however you define it, see my Comment 2.) of the gradient estimate. I don't see how this variance comes into the expression and the severity it has on the Bayes regret is not clear. Can you provide a discussion of this, if it is non-trivial? 2. You discuss the variability of the gradient estimate but don't define what you mean by it. The (usual) variance of the gradient estimate should be a matrix (variance-covariance) since the gradient itself is a vector. From a practical point of view, how do you compare variances in this case? Do you say that one is bigger than the other if the difference is a positive definite matrix? Or do you have another variance metric in mind? 3. It seems that you are measuring the convergence by looking at the norm of the gradient estimate and you base your loss function on this. What is your motivation for using this metric as a "convergence criterion"? Do you think you can obtain similar results by looking at other convergence metrics? 4. When subsampling the data as you describe in Section 3.2, there is another source of stochasticity introduced. Does the implied variance from this stochasticity have the same effect as the usual variance, i.e. the variance rising from that one has to estimate the expectation (even when not subsampling)? Do these variances affect your convergence metric (and hence your optimal weights) in different ways, or can we view them as coming from the same source of stochasticity? Minor comments: 1. Line 73: typo: "usually are usually lower". 2. Equations are part of sentences and should be terminated by full stop (when applicable) 3. You sometimes use small z and other times capital Z for what appears to be the same expressions. Please check for inconsistencies in the notation. 4. In many places, bayesian => Bayesian, i.e. capital B. 5. australian to Australian

Reviewer 3



The authors study variance reduction in black-box variational inference via control variates. Control variates are random variables of the same dimension as the stochastic gradient that have expectation zero; when added to the latter, the gradient’s expectation doesn’t change, but significant variance reduction can be achieved, accelerating convergence. The authors in particular analyze ensembles of different control variates and their weightings relative to the gradient. A Bayesian decision rule for optimally updating the control variates’ weights is presented. In contrast to prior work, their decision rule is regularized. Finally, experiments on a Bayesian logistic regression model are presented which confirm that using multiple control variance with the proposed weighting scheme lead to significant variance reduction and faster convergence, relative to using these control variates in isolation. The paper is very well written and very insightful. Besides its core contribution—-the control variate weighting scheme and decision rule—it also serves as a nice introduction to the topic of control variates for black box variational inference, which is an active area of research. One might worry that the main result of regularizing the weighting scheme by adding a small diagonal matrix to the empirical covariance of the control variates before inversion is minor, as Eq. 4 is well known. However, the derivation of this result is very interesting and sound, involving a Bayesian probabilistic model for the joint distribution of stochastic gradient and the control variates. I found this derivation very illuminating and relevant. A weakness are the experiments, which only cover logistic regression on small-scale data sets. However, here one may also argue that the experiments serve their purpose as to explain the phenomenon in the arguably simplest setup. Also, the authors investigated control variates which were not suggested and studied before, which is another contribution. Just one question: why did the authors use stochastic gradient with momentum in their experiments? Doesn’t this complicate the control variate story since the stochastic gradient is biased? All in all, the paper will certainly find its audience at NIPS. I enjoyed the paper and would recommend publishing it in its current form.